# Plasma Fibulin-5 Levels as an Independent Predictor of a Poor Outcome after an Aneurysmal Subarachnoid Hemorrhage

**DOI:** 10.3390/ijms232315184

**Published:** 2022-12-02

**Authors:** Yume Suzuki, Hiroki Oinaka, Hideki Nakajima, Mai Nampei, Fumihiro Kawakita, Yoichi Miura, Ryuta Yasuda, Naoki Toma, Hidenori Suzuki

**Affiliations:** Department of Neurosurgery, Mie University Graduate School of Medicine, 2-174 Edobashi, Tsu 514-8507, Japan

**Keywords:** early brain injury, delayed cerebral ischemia, extracellular matrix protein, fibulin, matricellular protein, prognostic prediction, subarachnoid hemorrhage, vasospasm

## Abstract

Aneurysmal subarachnoid hemorrhage (SAH) is a poor-outcome disease with a delayed neurological exacerbation. Fibulin-5 (FBLN5) is one of matricellular proteins, some of which have been involved in SAH pathologies. However, no study has investigated FBLN5’s roles in SAH. This study was aimed at examining the relationships between serially measured plasma FBLN5 levels and neurovascular events or outcomes in 204 consecutive aneurysmal SAH patients, including 77 patients (37.7%) with poor outcomes (90-day modified Rankin Scale 3–6). Plasma FBLN5 levels were not related to angiographic vasospasm, delayed cerebral ischemia, and delayed cerebral infarction, but elevated levels were associated with severe admission clinical grades, any neurological exacerbation and poor outcomes. Receiver-operating characteristic curves indicated that the most reasonable cut-off values of plasma FBLN5, in order to differentiate 90-day poor from good outcomes, were obtained from analyses at days 4–6 for all patients (487.2 ng/mL; specificity, 61.4%; and sensitivity, 62.3%) and from analyses at days 7–9 for only non-severe patient (476.8 ng/mL; specificity, 66.0%; and sensitivity, 77.8%). Multivariate analyses revealed that the plasma FBLN5 levels were independent determinants of the 90-day poor outcomes in both all patients’ and non-severe patients’ analyses. These findings suggest that the delayed elevation of plasma FBLN5 is related to poor outcomes, and that FBLN5 may be a new molecular target to reveal a post-SAH pathophysiology.

## 1. Introduction

About 85% of subarachnoid hemorrhage (SAH) cases are caused by ruptured intracranial aneurysms [1], and an aneurysmal SAH remains a life-threatening disease, due to a high risk of neurovascular complications, which usually occur within the first 2 weeks of onset [2]. Early-phase (from onset to day 3) complications include transient global cerebral ischemia, acute hydrocephalus, seizures, and stress-induced cardiomyopathy [3,4,5,6]. Late-phase (from day 4 to day 14 or later) complications include cerebral vasospasms and other types of delayed cerebral ischemia (DCI) [7]. In recent years, the concept of early brain injury (EBI) has become popular [8]. EBI develops within 3 days of the SAH onset, and the complex pathophysiology includes microthrombosis, early vasoconstriction, disturbance of the cerebral autoregulation, blood–brain barrier (BBB) disruption, venous drainage dysfunction, cortical spreading depolarization, and neuronal apoptosis [9], possibly followed by inflammatory reactions and a DCI [8].

Fibulin-5 (FBLN5) is a 66-kDa extracellular matrix protein (ECM) belonging to class II of the fibulin family [10], and is required for the construction of elastic fiber and angiogenesis [11,12,13]. It is reported that a lack of FBLN5 causes loose skin, aortic tortuosity, emphysema, and genital prolapse in vivo [11,12]. In contrast, FBLN5 was reported to prevent cortical cell apoptosis and the BBB disruption in a transient cerebral ischemia model of rats [14,15]. In a clinical setting, plasma FBLN5 levels were reported to increase in an acute phase of both ischemic and hemorrhagic strokes [16,17]. However, clinical significances of plasma FBLN5 levels have never been investigated in patients with an aneurysmal SAH. In this study, the authors measured the plasma FBLN5 levels serially, and examined whether plasma FBLN5 levels are related to the clinical course and outcomes in patients with an aneurysmal SAH.

## 2. Results

### 2.1. Clinical Variables Related to a Poor Outcome

In 305 consecutive patients registered in the period, the present study analyzed 204 patients, according to the inclusion and exclusion criteria (Figure 1). In five patients whose 90-day modified Rankin scales (mRSs) were missing, mRS at discharge was used instead: of those, four patients had poor outcomes (mRS 4 or 5), and one patient had a good outcome (mRS 2). In addition, the angiographic vasospasm was not evaluated in one patient due to their poor general condition, who was included in the study and analyzed, excluding the missing data.

The clinical variables of the SAH patients were shown in Table 1. The median age was 66.0 (interquartile range, 51.8–74.3) years old, and 51 patients (25.0%) were aged 75 years or older. The study population included 149 female patients (73.0%), 77 patients (37.7%) with World Federation of Neurological Surgeons (WFNS) grades IV-V at admission, 72 patients (35.3%) with modified a Fisher grade 4 at admission computed tomography (CT), and 67 patients (32.8%) with acute hydrocephalus. Most of ruptured aneurysms (186 patients, 91.2%) were located in the anterior circulation, and were treated with clipping (161 patients, 78.9%). Cerebrospinal fluid (CSF) drainage was performed in 69 patients (33.8%), and a Rho kinase inhibitor fasudil hydrochloride was prophylactically administered for a DCI in 201 patients (98.5%). As a result, an angiographic vasospasm, a DCI, and a delayed cerebral infarction occurred in 62 patients (30.4%), 29 patients (14.2%), and 45 patients (22.1%), respectively. Seventy-seven patients (37.7%) had poor outcomes (90-day mRS 3–6) and eight patients (3.9%) died.

The univariate analyses found that a 90-day poor outcome (mRS 3–6) was significantly associated with the following variables: increasing age, female, past history of cerebral infarction, non-pre-onset mRS 0, pre-onset mRS 1, WFNS grades IV-V at admission, non-modified Fisher grade 1, non-modified Fisher grade 3, modified Fisher grade 4, acute hydrocephalus, CSF drainage, DCI, and delayed cerebral infarction (Table 1).

### 2.2. Plasma FBLN5 Levels

The control samples were obtained from 10 patients with unruptured cerebral aneurysms, but one patient’s data were excluded because active gingivitis was associated and thereby the plasma FBLN5 levels were extremely high (899.0 ng/mL; Smirnov–Grubbs’s test, *p* = 0.03). Age was similar between the control and SAH patients (67.9 ± 9.5 versus 64.2 ± 14.6 years; *p* = 0.454; Mann–Whitney U test), and plasma FBLN5 levels were not different between male and female patients in the control (428.2 ± 38.7 versus 473.0 ± 91.2 ng/mL; *p* = 0.348; unpaired *t* test) and SAH (459.2 ± 197.8 versus 489.7 ± 209.8 ng/mL; *p* = 0.063; Mann–Whitney U test). Plasma FBLN5 levels tended to increase from days 1–3 to 10–12 after a SAH, but were not different between nine control patients and SAH patients: 448.1 ± 66.5 versus 438.1 ± 66.5 at days 1–3, 490.9 ± 220.3 at days 4–6, 486.1 ± 212.8 at days 7–9, and 510.8 ± 195.5 ng/mL at days 10–12 post-SAH (*p* = 0.469, 0.701, 0.603, and 0.368, respectively; Mann–Whitney U test; Figure 2a).

As to the relationships between the plasma FBLN5 levels and the severity of the SAH or the outcome measures, patients with WFNS grades IV-V at admission had significantly higher FBLN5 levels than those with WFNS grades I-III at days 4–6 and 7–9 post-SAH (Figure 2b; Appendix A). Plasma FBLN5 levels were not different among the modified Fisher grades (Figure 2c), and between patients with and without an angiographic vasospasm, a DCI, and a delayed cerebral infarction, at any sampling point (Figure 2d–f). However, plasma FBLN5 levels were significantly higher in poor-outcome (90-day mRS 3–6) patients, compared with good-outcome (90-day mRS 0–2) patients at days 4–6, 7–9, and 10–12 post-SAH (Figure 2g).

When compared between patients with and without any neurological exacerbation, including a DCI by day 14, compared to at admission, the plasma FBLN5 levels were significantly higher in the exacerbation group at days 7–9 post-SAH (Figure 2h). In the exacerbation group, the plasma FBLN5 levels tended to increase preceding the neurological exacerbation (Figure 2i).

### 2.3. Receiver-Operating Characteristic (ROC) Curve Analyses for the Performance of Plasma FBLN5, to Differentiate the Poor Outcomes

To differentiate the poor from the good outcomes, the ROC curve analyses indicated that plasma FBLN5 with a cut-off value of 458.2 ng/mL, resulted in a specificity of 62.2% and a sensitivity of 46.8% at days 1–3 post-SAH (area under the curve (AUC), 0.517; 95% confidence interval (CI), 0.435–0.599). As well, according to the curve, the specificity was 61.4% and the sensitivity was 62.3%, when a cut-off value of 487.2 ng/mL was set at days 4–6 (AUC, 0.627; 95% CI, 0.549–0.704); the specificity was 66.1% and the sensitivity was 54.6%, when a cut-off value of 499.4 ng/mL was set at days 7–9 (AUC, 0.622; 95% CI, 0.543–0.701); and the specificity was 89.8% and the sensitivity was 32.5%, when a cut-off value of 707.8 ng/mL was set at days 10–12 (AUC, 0.593; 95% CI, 0.511–0.676) (Appendix A). As the AUC was the highest at days 4–6, the plasma FBLN5 values at days 4–6 were used for the following multivariate analyses.

### 2.4. Independent Determinants for a 90-Day Poor Outcome

Among the significant variables on the univariate analyses (Table 1) and days 4–6 plasma FBLN5 cut-off values, the following variables had significant correlations: between the pre-onset mRSs 0 and 1 (r = −0.894), between WFNS grades I-III and IV-V (r = −1.000) at admission, between WFNS grades IV-V at admission, and the modified Fisher grade 4 (r = 0.335), between the modified Fisher grades 1 and 3 (r = −0.363), between the modified Fisher grades 3 and 4 (r = −0.717), between acute hydrocephalus and CSF drainage (r = 0.449), and between a DCI and a delayed cerebral infarction (r = 0.359). Among these similar clinical variables that were intercorrelated, only the variable with the smallest *p* value was used as a candidate variable for the multivariate analyses, in addition to other independent variables that were significantly related to a poor outcome on the univariate analyses. The multivariate analyses revealed that the plasma FBLN5 levels at days 4–6 post-SAH ≥ 487.2 ng/mL (odds ratio (OR), 2.55; 95% CI, 1.10-5.88; *p* = 0.029), were an independent determinant of a 90-day poor outcome, in addition to ≥75 years old, delayed cerebral infarction, non-pre-onset mRS 0, acute hydrocephalus, non-modified Fisher grade 3, and at admission, WFNS grades IV-V (Table 2).

In Appendix A, the representative cases were shown: the 90-day outcomes were poor when the plasma FBLN5 levels at days 4–6 post-SAH were ≥487.2 ng/mL, and good (mRS 0–2), when those were <487.2 ng/mL, regardless of the admission WFNS grades, the modified Fisher grades, and the presence or absence of a delayed cerebral infarction.

### 2.5. Analyses in Non-Severe SAH Cases

We also performed an analysis limited to patients with, at admission, WFNS grades I-III in which the diagnosis of a DCI is easier. The total number of cases was 127, of which 27 cases had poor outcomes (Appendix A). The univariate analyses were limited to non-severe SAH cases, although most of the 90-day poor outcome-related variables were similar to the analyses in all patients, a pre-onset mRS 1, non-modified Fisher grades 1 and 3, and CSF drainage were no longer significant variables, and a past history of hypertension, a WFNS grade I not at admission, a WFNS grade III at admission, a ruptured aneurysm located at the anterior cerebral artery (ACA) and internal carotid artery (ICA), were added as 90-day poor outcome-related variables (Table 1 and Appendix A). The relationships between the plasma FBLN5 levels and the outcome measures were also similar, but patients with a DCI had higher values at days 7–9 post-SAH in the analyses limited to the non-severe SAH cases (Figure 3 and Appendix A). The ROC curve analyses for the performance of the plasma FBLN5 levels, in order to differentiate the 90-day poor from the good outcomes, showed a better AUC, compared with the analyses of all patients, especially at days 7–9 post-SAH: a cut-off value of 476.8 ng/mL, a sensitivity of 77.8%, and a specificity of 66.0% (AUC, 0.749; 95% CI, 0.642–0.856; Appendix A). All significant variables on the univariate analyses and a cut-off value of plasma FBLN5 at days 7–9 post-SAH were candidate variables for the multivariate analyses, but only the variable with the smallest *p* value was used among the following intercorrelated variables: between ACA and ICA (r = −0.644), between DCI and delayed cerebral infarction (r = 0.506), and between an at admission WFNS grade III and acute hydrocephalus (r = 0.328). The multivariate analyses revealed that the plasma FBLN5 levels at days 7–9 post-SAH ≥ 476.8 ng/mL (OR, 18.14; 95% CI, 3.25–101.39; *p* < 0.001) were an independent determinants of a 90-day poor outcome, although some independent variables were different from those in the analyses for all SAH patients (Appendix A).

## 3. Discussion

The novel findings in this study were as follows: (1) the plasma FBLN5 levels in patients with WFNS grades IV-V at admission, were significantly higher than those in WFNS grades I-III patients, at days 4–9 post-SAH; (2) the plasma FBLN5 levels were not related to an angiographic vasospasm, DCI, or delayed cerebral infarction, but any neurological exacerbation and 90-day poor outcome were associated with elevated FBLN5 levels at days 7–9 and days 4–12, respectively, in all SAH patients’ analyses; (3) in patients with WFNS grades I-III at admission, higher plasma FBLN5 levels were not related to an angiographic vasospasm and delayed cerebral infarction, but those at days 7–9 and days 1–12 were associated with a DCI and any neurological exacerbation or 90-day poor outcome, respectively; and (4) higher plasma FBLN5 levels were independent determinants of a 90-day poor outcome at days 4–6 in all patients’ analyses, and at days 7–9 in the non-severe patients’ analyses. These findings suggest that the plasma FBLN5 levels can be a predictor of the 90-day outcomes after a SAH, and that FBLN5 may be related to an EBI and a delayed neurological exacerbation, leading to poor outcomes.

FBLN5 is one of ECMs belonging to the FBLN family, and it is also a matricellular protein [10]. FBLNs have a repeated calcium binding epidermal growth factor (EGF)-like motifs and a C-terminal fibulin domain [18]. FBLN5 is a family of the short type FBLNs (named as class II type FBLNs) [10], and reported to play an important role in the development of elastic fibers in vivo [18]. FBLN5 is also known as an embryonic vascular EGF-like repeat-containing protein (EVEC) [19] or developing arteries and neural crest EGF-like (DANCE) [20,21], and is expressed mainly in the vascular smooth muscle cells of the developing arteries in embryos [19,22]. Its expression is downregulated in adult arteries, except for tissues with ongoing angiogenesis [19], but the re-expression is observed in balloon-injured vessels or atherosclerotic lesions [20]. Thus, FBLN5 has been involved in the vascular remodeling of injured or atherosclerotic vessels through the growth regulation, matrix maturation, or signaling between vascular smooth muscles and endothelial cells [19,20]. In SAH, FBLN5 may increase the reflecting of the rupture of intracranial aneurysms or vessel injuries, associated with an angiographic vasospasm, but the findings in this study did not support the hypothesis.

In the central nervous system, there are only several studies reported as to FBLN5. Guo et al. reported that overexpressed FBLN5 reduced the reactive oxygen species production, the apoptosis of cortical cells and capillary endothelial cells, and the BBB disruption, resulting in improved neurological functions, in a transient cerebral ischemia model of rats, although the infarct volume was not decreased [14]. Zang et al. reported that limb remote ischemic postconditioning reduced the infarct volume, improved the neurobehavior, and inhibited the BBB disruption by increasing FBLN5 in rat transient cerebral ischemia models [15]. In a clinical setting, Hu et al. reported that the plasma FBLN5 levels significantly increased within 3 days after an intracerebral hemorrhage, and were correlated with the disease severity and the 90-day outcomes [16]. According to a report by Elshony et al., the plasma FBLN5 levels were increased in all types of strokes, especially in hemorrhagic strokes, in the first few hours of onset, and the higher values were observed in more severe diseases [17]. Taken together, the experimental studies have shown that FBLN5 is neuroprotective for ischemic stroke, whereas clinical studies have found that higher plasma FBLN5 levels in an acute phase are associated with more severe ischemic or hemorrhagic strokes with worse outcomes. Although the present study is the first report to investigate the plasma FBLN5 levels serially from an acute phase to a subacute phase, the findings were similar to the previous clinical studies, in terms of that higher plasma FBLN5 levels at a certain time were associated with worse admission clinical grades and outcomes. However, it was unique that higher plasma FBLN5 levels in a delayed phase (FBLN5 levels at days 4–6 in all patients’ analyses and those at days 7–9 in non-severe patients’ analyses) were independent determinants for 90-day poor outcomes.

The matricellular proteins, such as tenascin-C, galectin-3, periostin, secreted protein acidic, and rich in cysteine, and osteopontin (OPN), have been implicated in diverse post-SAH pathologies [23,24,25,26,27,28,29,30,31]. Most of the matricellular proteins have been reported to exert harmful effects on post-SAH pathologies, such as neuroinflammation [24], BBB disruption [23,26,30,31], neuronal apoptosis [24], and cerebral vasospasms [29]. Moreover, only OPN has been repeatedly reported to be protective against an EBI, in terms of neuronal apoptosis [27], BBB disruption [27,32], and microcirculatory dysfunctions [33,34], as well as cerebral vasospasms [28] in experimental animal models. In a clinical setting, however, even OPN has been related to poor outcomes: that is, the plasma OPN levels increased after an aneurysmal SAH, and were significantly higher in patients with 90-day poor outcomes, compared with good outcome patients [35]. Although FBLN5 is one of the matricellular proteins, the functional significance has never been investigated in experimental SAH models. Considering the findings of the basic studies on FBLN5 in other fields, as described above, FBLN5 may act protectively similar to OPN. In the present clinical study, the higher plasma FBLN5 levels were associated with poor outcomes: the findings are also similar to those in clinical studies of OPN in patients with an aneurysmal SAH [35,36]. At the moment, thus, the authors hypothesize that the plasma FBLN5 levels may increase in a delayed fashion, reflecting the severity of an EBI that can be indicated by the WFNS grades, and that the increased FBLN5 may act protectively against an EBI but be insufficient to recover from the EBI, followed by a secondary neurological exacerbation and poor outcomes.

This study has some limitations. First, it is unknown if the sampling point in this study was the best, because the pathophysiology of SAH patients may change individually. Second, this study was a multicenter study, in which magnetic resonance imaging was not required in an acute phase of a SAH, and neurovascular events were diagnosed by CT. Third, it was not tested if FBLN5 is superior to the previously reported ones, as a biomarker. This is because there are no established or validated biomarkers known in a SAH, although many plausible biomarkers have been reported, including non-specific but central nervous system-derived cell death, or recovery-related ones, such as neurofilaments and S100β [37]. In addition, this study focused on discovering a new molecular target to elucidate the pathology of a SAH, rather than establishing a new biomarker: in the first place, we think that a single biomarker may not adequately capture the multiple complex and inter-related pathophysiological cascades associated with a SAH. Fourth, the authors cannot determine where FBLN5 was produced, because only the plasma FBLN5 levels were measured, and the CSF FBLN5 levels were not measured. However, this is the first study to show that FBLN5 may be involved in the post-SAH pathology. Future experimental and clinical studies would clarify the function, mechanisms and clinical significance of FBLN5.

## 4. Materials and Methods

The Institutional Ethics Committee approved the study (approval No. 2544), and the written informed consent was obtained from the relatives.

### 4.1. Study Population

The present study used the data of consecutive patients registered in the prospective registry for searching mediators of neurovascular events after an aneurysmal SAH (pSEED), which was conducted in nine tertiary referral centers in Mie prefecture, in Japan, from September 2013 to December 2016 [35,38,39,40,41,42]. In the registry, the clinical variables were recorded, and the plasma samples were serially collected. The inclusion criteria of the registry were as follows: ≥20 years old at onset, mRS 0–2 before onset, the SAH was diagnosed by CT scans, a saccular aneurysm as the cause of the SAH was confirmed on the CT angiography or digital subtraction angiography, and an aneurysmal obliteration by clipping or coil embolization occurred within 48 h of onset. The following was excluded from the present study (Figure 1): unknown etiology of the SAH, a SAH caused by a ruptured fusiform, dissecting, traumatic, mycotic, or arteriovenous malformation-related aneurysms, treatment by parent artery occlusion with or without bypass surgery, no prophylactic medication for a DCI (fasudil hydrochloride or cilostazol), any angiographic or treatment-related complications, concomitant inflammatory diseases that are known to upregulate FBLN5, missing blood samples, and missing data of the clinical variables. Timing, treatment modality (clipping or coil embolization) of the aneurysmal obliteration, and other medical management or treatment strategies, depended on the on-site investigators. In the present study, the plasma FBLN5 levels were measured using stocked plasma samples, and the relationship between the plasma FBLN5 levels and the clinical variables was retrospectively analyzed.

### 4.2. Clinical Variables

The following variables were analyzed: age, sex, past history, co-morbidities (hypertension, dyslipidemia, and diabetes mellitus; previously diagnosed or on medication), current smoking, family history of SAH, pre-onset mRS, location of the ruptured aneurysms, the WFNS grade at admission, the modified Fisher grade at admission, CT scans [43], and acute hydrocephalus. Age was divided into two groups, over and under 75 years old [42]. In the analyses of all SAH cases, the WFNS grades at admission were divided into non-severe cases (WFNS grades I-III) and severe cases (WFNS grades IV-V) [38], and in the analyses of the non-severe cases, the WFNS grades were analyzed individually [38]. Acute hydrocephalus was recorded when there was evidence of ventriculomegaly, proved by CT scans at admission, which was considered to cause neurological impairments, such as disturbance of consciousness. In the case of acute hydrocephalus, a catheter for CSF drainage was placed at the clipping or immediately before or after the coil embolization. As the treatment-related variables, the following factors were evaluated: treatment modality (clipping or coil embolization) performed for the aneurysmal obliteration, CSF drainage, including ventricular, cisternal, and spinal drainage, to manage the hydrocephalus and/or to promote the hematoma clearance, medication to prevent a DCI (intravenously administered fasudil hydrochloride, oral or enteral cilostazol, ethyl icosapentate, and statin). The outcomes were assessed as to the angiographic vasospasm, DCI, delayed cerebral infarction, and mRS, at 90 days post-SAH. A DCI was defined as otherwise an unexplained clinical deterioration (i.e., focal neurological impairments, a decrease of at least two points on the Glasgow coma scale, or both), which lasted for at least one hour [2]. Other potential causes of clinical deterioration were excluded, based on clinical assessments including CT scans, magnetic resonance images, electroencephalography, and laboratory studies. Any clinical deterioration including, DCI by day 14, compared with at admission, was also recorded as a neurological exacerbation, irrespective of the causes. An angiographic vasospasm was defined as a 50% or more reduction in the baseline vessel diameter of the major cerebral arteries on a CT angiography or digital subtraction angiography, irrespective of the symptoms, which was performed around 7 days post-SAH, and a neurological deterioration. A delayed cerebral infarction was defined as newly developed non-iatrogenic infarcts, which were not observed on CT scans or magnetic resonance images on the day after the aneurysmal obliteration, and developed within 1 month of the SAH. In cases where the 90-day mRS was missing, the mRS at discharge was used instead. The determination of these variables was made at each center, and the organizing committee qualified them.

### 4.3. Measurement of Plasma FBLN5

Following the aneurysmal obliteration, the blood samples were serially collected with minimal stasis, from a peripheral vein at days 1–3, 4–6, 7–9, and 10–12 after onset. All samples were centrifuged for 5 min at 3000× *g*, and the supernatants were stored at −78 °C, until assayed. The plasma FBLN5 levels were determined using a commercially available enzyme-linked immunosorbent assay kit for human FBLN5 (27121; IBL, Gunma, Japan). As a control, the blood samples were also obtained from 10 patients with unruptured cerebral aneurysms who provided written informed consent before any invasive procedure, and the plasma FBLN5 levels were determined, as described above.

### 4.4. Statistical Analysis

All statistical analyses were performed with SPSS software, version 28.0.1.0 (IBM, Armonk, New York, NY, USA). The continuous variables were expressed as a mean ± standard deviation (SD) or standard error of the mean (for graphs), or a median ±25–75 percentile, and were compared between two groups using unpaired *t* tests or Mann-Whitney U tests, respectively, and among more than three groups, by a one-way analysis of variance or Kruskal-Wallis test with Steel-Dwass’s post hoc test, as appropriate, after examining if each population compared, followed a normal distribution using Shapiro-Wilk tests. The categorical variables were reported as percentages and were compared using Pearson’s chi-squared tests or Fisher’s exact tests, as appropriate. The ROC curves and the AUC were analyzed to obtain the cut-off values of the plasma FBLN5 for the 90-day poor outcome. The correlation between the plasma FBLN5 levels and the clinical variables, or between the clinical variables was evaluated by Pearson’s correlation coefficient for continuous variables or Spearman’s correlation coefficient for categorical variables. The correlation coefficient |r| > 0.3 was considered a significant correlation. Any variable with a *p* value of <0.05 on the univariate analyses was entered into the multivariate logistic regression models using the 90-day dichotomous mRS outcome (good or poor), as the dependent variable, although only the variable with the smallest *p* value was used as a candidate variable, among the similar clinical variables that were intercorrelated. The multivariate logistic regression analyses were performed by the forward stepwise method. The analyses were performed in all patients, as well as in non-severe SAH (defined as WFNS grades I-III at admission) patients. A significant level was set at a *p* value < 0.05.

## 5. Conclusions

This study first revealed that the delayed elevation of the plasma FBLN5 levels was an independent determinant for 90-day poor outcomes, and that FBLN5 might be a new molecular target to clarify the post-SAH pathologies and to develop a new therapy against them.

## Figures and Tables

**Figure 1 ijms-23-15184-f001:**
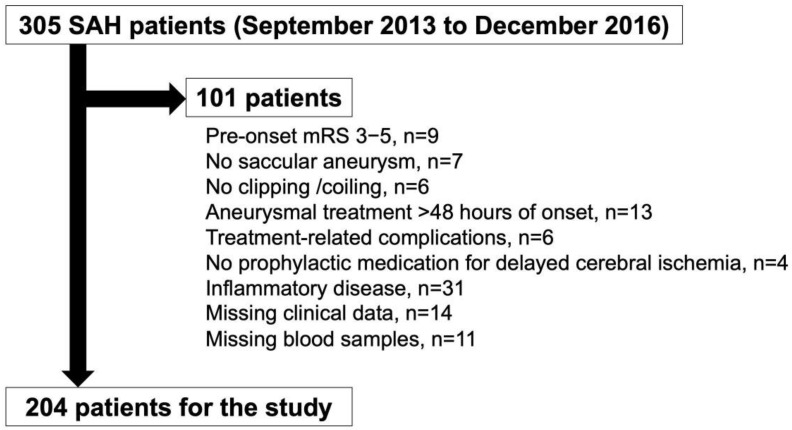
A flow chart showing the included and excluded patients in this study. mRS, modified Rankin scale; SAH, subarachnoid hemorrhage.

**Figure 2 ijms-23-15184-f002:**
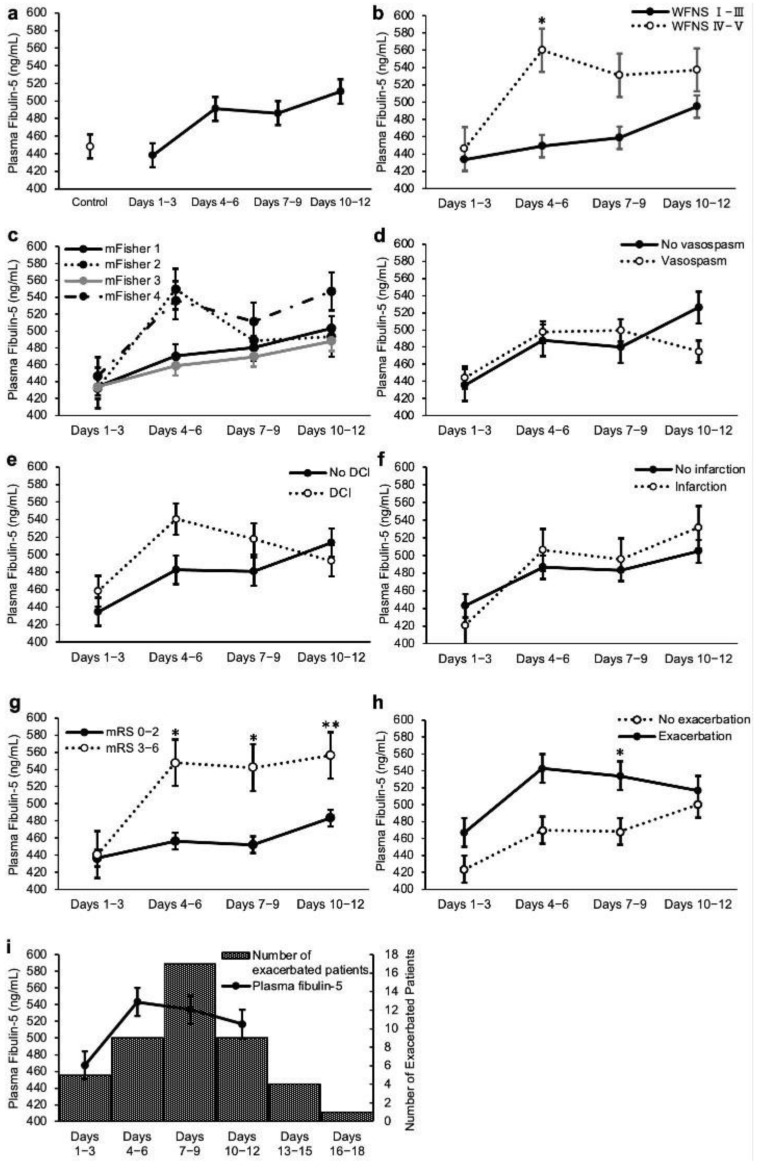
Chronological changes in the plasma fibulin-5 levels after a subarachnoid hemorrhage (SAH). Comparison between the control and SAH patients (**a**), between SAH patients with World Federation of Neurological Surgeons (WFNS) grades I-III and IV-V at admission (**b**), among the modified Fisher grades (**c**), between patients with and without angiographic vasospasm (**d**), delayed cerebral ischemia (DCI); (**e**), or delayed cerebral infarction (**f**), between the 90-day modified Rankin scale (mRS) 0–2 and 3–6 (**g**), and between patients with and without neurological exacerbations (**h**). Relationships between the changes in the plasma fibulin-5 levels and the timing of the neurological exacerbation (**i**). Data, a mean ±standard error of the mean. * *p* < 0.05, Mann–Whitney U test; ** *p* < 0.05, unpaired *t* test.

**Figure 3 ijms-23-15184-f003:**
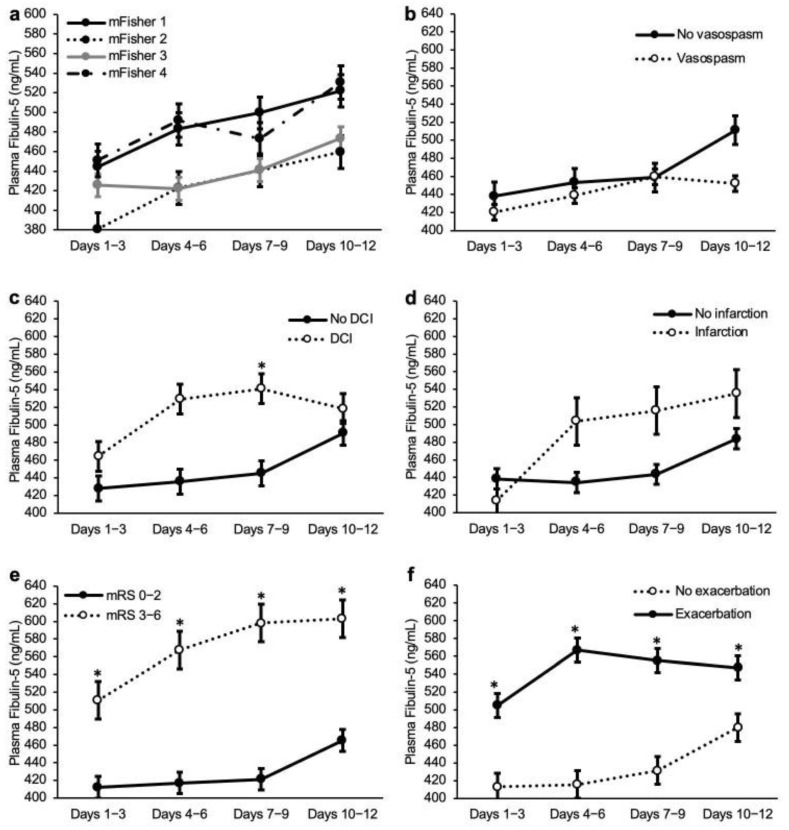
Chronological changes in the plasma fibulin-5 levels in patients with, at admission, World Federation of Neurological Surgeons grades I-III of a subarachnoid hemorrhage. Comparison among the modified Fisher grades (**a**), between the patients with and without an angiographic vasospasm (**b**), a delayed cerebral ischemia (DCI); (**c**), or a delayed cerebral infarction (**d**), between the 90-day modified Rankin scale (mRS) 0–2 and 3–6 (**e**), and between patients with and without a neurological exacerbation (**f**). Data, a mean ±standard error of the mean. * *p* < 0.05, Mann-Whitney U test.

**Table 1 ijms-23-15184-t001:** Clinical variables related to the 90-day poor outcome or modified Rankin scale (mRS) 3–6.

	Total (n = 204)	mRS 0–2 (n = 127)	mRS 3–6 (n = 77)	*p* Value
Age				
Median (IQR), years	66.0 (51.8–74.3)	60.0 (48.0–69.0)	74.0 (66.0–82.0)	<0.001 ^a^
≥75 years old	51 (25.0)	14 (11.0)	37 (48.1)	<0.001 ^b^
Female	149 (73.0)	86 (67.7)	63 (81.8)	0.028 ^b^
Past history				
SAH	10 (4.9)	5 (3.9)	5 (6.5)	0.308 ^c^
Cerebral infarction	9 (4.4)	2 (1.6)	7 (9.1)	0.016 ^c^
Hypertension	90 (44.1)	56 (44.1)	34 (44.2)	0.993 ^b^
Dyslipidemia	23 (11.3)	12 (9.4)	11 (14.3)	0.290 ^b^
Diabetes mellitus	15 (7.4)	8 (6.3)	7 (9.1)	0.459 ^b^
Family history of SAH	21 (10.3)	16 (12.6)	5 (6.5)	0.164 ^b^
Current smoking	45 (22.1)	33 (26.0)	12 (15.6)	0.082 ^b^
Pre-onset mRS				
0	188 (92.2)	123 (96.9)	65 (84.4)	0.002 ^c^
1	13 (6.4)	4 (3.1)	9 (11.7)	0.018 ^c^
2	3 (1.5)	0 (0.0)	3 (3.9)	0.052 ^c^
Admission WFNS grade				
I-III	127 (62.3)	100 (78.7)	27 (35.1)	<0.001 ^b^
IV-V	77 (37.7)	27 (21.3)	50 (64.9)	<0.001 ^b^
Modified Fisher grade				
1	25 (12.3)	22 (17.3)	3 (3.9)	0.003 ^c^
2	8 (3.9)	7 (5.5)	1 (1.3)	0.127 ^c^
3	99 (48.5)	74 (58.3)	25 (32.5)	<0.001 ^b^
4	72 (35.3)	24 (18.9)	48 (62.3)	<0.001 ^b^
Acute hydrocephalus	67 (32.8)	25 (19.7)	42 (54.5)	<0.001 ^b^
Ruptured AN location				
ACA	70 (34.3)	48 (37.8)	22 (28.6)	0.179 ^b^
ICA	76 (37.3)	45 (35.4)	31 (40.3)	0.489 ^b^
MCA	40 (19.6)	23 (18.1)	17 (22.1)	0.489 ^b^
VA-BA	18 (8.8)	11 (8.7)	7 (9.1)	0.917 ^b^
Anterior circulation	186 (91.2)	116 (91.3)	70 (90.9)	0.917 ^b^
Multiple Aneurysms	56 (27.5)	34 (26.8)	22 (28.6)	0.780 ^b^
Treatment modality				
Clipping	161 (78.9)	96 (75.6)	65 (84.4)	0.134 ^b^
Coiling	43 (21.1)	31 (24.4)	12 (15.6)	0.134 ^b^
CSF drainage	69 (33.8)	35 (27.6)	34 (44.2)	0.015 ^b^
Prophylactic drug for DCI				
Fasudil hydrochloride	201 (98.5)	126 (99.2)	75 (97.4)	0.319 ^c^
Cilostazol	166 (81.4)	105 (82.7)	61 (79.2)	0.539 ^b^
Ethyl icosapentate	103 (50.5)	68 (53.5)	35 (45.5)	0.263 ^b^
Statin	50 (24.5)	35 (27.6)	15 (19.5)	0.193 ^b^
Angiographic vasospasm	62 (30.4)	33 (26.0)	29 (37.7)	0.068 ^b^
DCI	29 (14.2)	11 (8.7)	18 (23.4)	0.004 ^b^
Delayed cerebral infarction	45 (22.1)	15 (11.8)	30 (39.0)	<0.001 ^b^

Data are number of cases (%) unless otherwise specified and compared between mRS 0–2 and 3–6. *p* values are determined by ^a^ Mann–Whitney U test, ^b^ Pearson’s chi-squared test, or ^c^ Fisher’s exact test. ACA, anterior cerebral artery; AN, aneurysm; CSF, cerebrospinal fluid; DCI, delayed cerebral ischemia; ICA, internal carotid artery; IQR, interquartile range; MCA, middle cerebral artery; SAH, subarachnoid hemorrhage; VA-BA, vertebral artery-basilar artery; WFNS, World Federation of Neurological Surgeons.

**Table 2 ijms-23-15184-t002:** Multivariate logistic regression analyses for predicting a 90-day poor outcome after a subarachnoid hemorrhage.

Variables	Odds Ratio	95% CI	*p* Value
Age ≥ 75 years old	3.31	1.38–7.93	0.007
Delayed cerebral infarction	12.64	4.48–35.68	<0.001
Fibulin-5 at days 4–6 ≥ 487.2 ng/mL	2.55	1.10–5.88	0.029
Pre-onset mRS 0	0.10	0.02–0.62	0.013
Acute hydrocephalus	4.84	2.03–11.54	<0.001
Modified Fisher grade 3	0.30	0.13–0.71	0.006
Admission WFNS grades IV-V	5.93	2.61–13.47	<0.001
Female			0.887
Past history of a cerebral infarction			0.120

Variables are selected by the forward stepwise method. Plasma fibulin-5 levels at days 4–6 are categorized using the cut-off value that was determined in Appendix A. CI, confidence interval; mRS, modified Rankin scale; WFNS, World Federation of Neurosurgical Surgeons.

## Data Availability

Data from this study will be made available to qualified investigators upon reasonable inquiry.

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
