# Peer review of "Plasma Fibulin-5 Levels as an Independent Predictor of a Poor Outcome after an Aneurysmal Subarachnoid Hemorrhage"

_ijms, 2022, doi:10.3390/ijms232315184_

Round 1

Reviewer 1 Report

The current manuscript while interesting appears to be biased and lacks scientific rigor. The study must include equal number of age matched male and female patients and healthy controls. The authors must directly compare FBLN5 levels with previously established biomarkers to demonstrate superiority of FBLN5. The authors must provide de-identified SEBES score and representative MRI data for the healthy subjects as well as male and female patients. It would be best to compare both plasma as well as CSF FBLN5 levels in healthy subjects and the SAH patients at different time points to rigorously validate the data.

Author Response

We would like to thank you for your insightful comments. We have improved our manuscript according to your suggestions, and the modified sentences or phrases in our manuscript are shown in red.

The current manuscript while interesting appears to be biased and lacks scientific rigor. The study must include equal number of age matched male and female patients and healthy controls.

Thank you for your suggestions. However, this study did not focus on comparisons between healthy controls and SAH patients, but focused on the relationship between poor outcomes and plasma FBLN5 levels in SAH patients. Multivariate analyses using variables including age, sex and plasma FBLN5 levels revealed that plasma FBLN5 level was an independent determinant for poor outcomes after SAH (Table 2 and Supplementary Table S4). According to your suggestions, however, we added the following description on lines 96 to 100: “Age was similar between control and SAH patients (67.9 ±9.5 versus 64.2 ±14.6 years; p=0.454; Mann-Whitney U test), and plasma FBLN5 levels were not different between male and female patients in control (428.2 ±38.7 versus 473.0 ±91.2ng/ml; p =0.348; unpaired t test) and SAH (459.2 ±197.8 versus 489.7 ±209.8ng/ml; p =0.063; Mann-Whitney U test).” Unpaired t tests were used when each population being compared followed a normal distribution using Shapiro-Wilk tests, and otherwise Mann-Whitney U tests were used.

The authors must directly compare FBLN5 levels with previously established biomarkers to demonstrate superiority of FBLN5.

Thank you for your suggestions. However, this study did not focus on the usefulness of FBLN5 as a biomarker, but was performed to examine whether it can be a new molecular target. According to your suggestions, we modified the sentences in the Abstract section on lines 25 to 27 as follows: “These findings suggest that delayed elevation of plasma FBLN5 is related to poor outcomes, and that FBLN5 may be a new molecular target to reveal post-SAH pathophysiology.” We also added the Conclusions section on lines 359 to 363 as follows: “5. Conclusions: This study first revealed that delayed elevation of plasma FBLN5 levels was an independent determinant for 90-day poor outcomes, and that FBLN5 might be a new molecular target to clarify post-SAH pathologies and to develop a new therapy against them.”

The authors must provide de-identified SEBES score and representative MRI data for the healthy subjects as well as male and female patients.

Thank you for your suggestions. As you suggested, SEBES score is a radiographic marker of early brain injury, but this study focused on neither early brain injury nor biomarker discovery. In addition, this study was a multicenter collaborative study, and there were not many facilities that performed MRI examinations in an acute phase of SAH. According to your suggestions, we added the following sentences as one of limitations of this study on lines 266 to 268: “Second, this study was a multicenter study, in which magnetic resonance imaging was not required in an acute phase of SAH, and neurovascular events were diagnosed by CT.”

It would be best to compare both plasma as well as CSF FBLN5 levels in healthy subjects and the SAH patients at different time points to rigorously validate the data.

Thank you for your suggestions. We agree with you in this regard. Therefore, we described this point as one of limitations on lines 268 to 269. However, we believe that this paper is significant in that this is the first study to reveal that FBLN5 may be involved in post-SAH pathology. We hope that this reviewer understands our thoughts.

Reviewer 2 Report

Dear Authors,

I am glad to have the opportunity to review your work. This study was aimed to examine the relationships between serially measured plasma FBLN5 levels and 15 neurovascular events or outcomes in 204 consecutive aneurysmal SAH patients including 77 pa-16 tients (37.7%) of poor outcomes (90-day modified Rankin Scale 3–6).

The study has great design, excellent statistical analysis and good presentation of the results.

However, my objective is that the section Materials and methods needs to be written before the Results and the Discussion, and not at the end of the Manuscript. Also, Conclusion needs to be added.

Therefore, I recommend minor revision of the paper.

Author Response

We would like to thank you for your encouraging comments. We have improved our manuscript according to your suggestions, and the modified sentences or phrases in our manuscript are shown in red.

I am glad to have the opportunity to review your work. This study was aimed to examine the relationships between serially measured plasma FBLN5 levels and neurovascular events or outcomes in 204 consecutive aneurysmal SAH patients including 77 patients (37.7%) of poor outcomes (90-day modified Rankin Scale 3–6). The study has great design, excellent statistical analysis and good presentation of the results. However, my objective is that the section Materials and methods needs to be written before the Results and the Discussion, and not at the end of the Manuscript. Also, Conclusion needs to be added. Therefore, I recommend minor revision of the paper.

Thank you for your suggestions. The order of the section “Materials and Methods” followed the submission guidelines of this journal. According to your suggestions, we added the Conclusions section on lines 359 to 363 as follows: “5. Conclusions: This study first revealed that delayed elevation of plasma FBLN5 levels was an independent determinant for 90-day poor outcomes, and that FBLN5 might be a new molecular target to clarify post-SAH pathologies and to develop a new therapy against them.”

Round 2

Reviewer 1 Report

The authors must provide representative de-identified CT data to validate their research findings. Since the authors are trying to establish Fibulin-5 levels as an independent predictor of poor 2 outcome after aneurysmal subarachnoid hemorrhage, it would be necessary to compare the expression levels of Fibulin-5 with well established markers such as Neurofilament light chain or S100A8 in the plasma of healthy controls and SAH patients. All of the supplementary data can become part of the main manuscript. 

Author Response

    We would like to thank you for your insightful comments. We have improved our manuscript according to your suggestions, and the modified sentences or phrases in our manuscript are shown in red.

The authors must provide representative de-identified CT data to validate their research findings.

    Thank you for your important suggestions. According to your suggestions, we added the supplemental figure S2 to show 3 representative cases, and the following sentences on lines 155 to 158: ‘‘In Supplementary Fig. S2, representative cases were shown: the 90-day outcomes were poor when plasma FBLN5 levels at days 4−6 post-SAH were ≥487.2ng/ml, and good (mRS 0–2) when those were <487.2ng/ml, regardless of admission WFNS grades, modified Fisher grades, and the presence or absence of delayed cerebral infarction.’’

Since the authors are trying to establish Fibulin-5 levels as an independent predictor of poor 2 outcome after aneurysmal subarachnoid hemorrhage, it would be necessary to compare the expression levels of Fibulin-5 with well established markers such as Neurofilament light chain or S100A8 in the plasma of healthy controls and SAH patients. All of the supplementary data can become part of the main manuscript. 

     Thank you for your suggestions. As you suggested, S100 protein and neurofilament light chain are potentially useful biomarkers for central nervous system diseases associated with cell death and recovery, but even these biomarkers have never been validated in SAH. Therefore, we did not compare plasma levels of fibulin-5 with those of other biomarkers, and that is a limitation of this study. According to your suggestions, we added one reference (reference #37) and the following sentences as one of limitations in this study on lines 271 to 278: “Third, it was not tested if FBLN5 is superior to previously reported ones as a biomarker. This is because there are no established or validated biomarkers known in SAH, although many plausible biomarkers have been reported, including non-specific but central nervous system-derived cell death or recovery-related ones such as neurofilaments and S100β [37]. In addition, this study focused on discovering a new molecular target to elucidate the pathology of SAH, rather than establishing a new biomarker: in the first place, we think that a single biomarker may not adequately capture the multiple complex and inter-related pathophysiological cascades associated with SAH.’’